# Two Heads Are Enough: DualU-Net, a Fast and Efficient Architecture for Cell Classification and Segmentation

**David Anglada-Rotger** [iD]                          DAVID.ANGLADA@UPC.EDU
**Berta Jansat**                                        BERTA.JANSAT@UPC.EDU
**Ferran Marques**                                    FERRAN.MARQUES@UPC.EDU
**Montse Pardàs**                                     MONTSE.PARDAS@UPC.EDU
*Image Processing Group (GPI), Universitat Politècnica de Catalunya (UPC), Barcelona, Spain*

**Editors:** Accepted for publication at MIDL 2025

## Abstract

Accurate detection and classification of cell nuclei in histopathological images are critical for both clinical diagnostics and large-scale digital pathology workflows. In this work, we introduce DualU-Net, a fully convolutional, multi-task architecture designed to streamline cell nuclei classification and segmentation. Unlike the widely adopted three-decoder paradigm of HoVer-Net, DualU-Net employs only two output heads: a segmentation decoder that predicts pixel-wise classification maps and a detection decoder that estimates Gaussian-based centroid density maps. By leveraging these two outputs, our model effectively reconstructs instance-level segmentations. The proposed architecture results in significantly faster inference, reducing processing time by up to ×5 compared to HoVer-Net, while achieving classification and detection performance comparable to state-of-the-art models. Additionally, our approach demonstrates greater computational efficiency than CellViT and NuLite. We further show that DualU-Net is more robust to staining variations, a common challenge in digital pathology workflows. The model has been successfully deployed in clinical settings as part of the DigiPatICS initiative, operating across eight hospitals within the Institut Català de la Salut (ICS) network, highlighting the practical viability of DualU-Net as an efficient and scalable solution for nuclei segmentation and classification in real-world pathology applications. The code and pretrained model weights are publicly available on https://github.com/davidanglada/DualU-Net.

**Keywords:** Cell Nuclei Classification, Cell Nuclei Segmentation, Digital Pathology, MultiTask Learning, Deep Learning, Computational Efficiency

## 1. Introduction

Digital pathology, powered by AI, is revolutionizing cancer diagnosis by automating cell detection and classification (Song et al., 2023). However, computational efficiency and robustness remain key challenges for real-world deployment. Pathologists analyze Whole Slide Images (WSIs) across multiple histological stains, such as Hematoxylin and Eosin (H&E) and immunohistochemical markers like Ki-67, to assess tumor characteristics. This manual process is time-consuming and subject to interobserver variability (Corona et al., 1996; Dano et al., 2020), making automated solutions essential for improving efficiency and consistency in clinical workflows.

Cell detection and classification are fundamental tasks in computational pathology, as accurate quantification of different cell types informs diagnostic and prognostic decisions. While segmentation aids visualization, classification remains the primary clinical objective.

Convolutional Neural Networks (CNNs) are widely used for these tasks, with U-Net (Ronneberger et al., 2015) being a popular choice due to its encoder-decoder structure and skip connections. However, U-Net struggles with overlapping and clustered cells, leading to the development of more advanced models like HoVer-Net (Graham et al., 2019), which employs a three-decoder architecture: one for binary segmentation, another for horizontal-vertical (HV) vector field prediction to separate clustered cells, and a third for cell classification. While this multi-task approach has been widely adopted (Hörst et al., 2024; Tommasino et al., 2024; Chen et al., 2025), maintaining three decoder heads increases computational cost and inference time, limiting clinical feasibility.

Beyond segmentation and classification, stain variability poses an additional challenge. Differences in staining protocols, scanner settings, and tissue preparation introduce significant variations across datasets, affecting model generalization. Robust models must be resilient to these variations to ensure reliable performance across different laboratories.

In this paper, we propose *DualU-Net*, a streamlined deep learning architecture for cell classification and segmentation across multiple histological stains. Our primary contribution is demonstrating that *two decoder heads are enough*, challenging the need for HoVer-Net's three-decoder scheme. We dispense with the binary segmentation branch and we replace the HV vector field branch with a Gaussian-based centroid estimation approach. Our key contributions include: *i)* a dual-decoder architecture proving that two heads are sufficient for cell detection, classification and segmentation in multiple stains; *ii)* comparable classification and detection performance, aligning with pathologists' focus on cell quantification over precise segmentation contours; *iii)* fast and efficient inference, making the model suitable for real-time clinical deployment; *iv)* robustness to stain variations, ensuring consistent performance across different histological markers; and *v)* real-world deployment, with DualU-Net integrated into the DigiPatICS project (Temprana-Salvador et al., 2022) and deployed across eight hospitals within the Institut Català de la Salut (ICS) of Catalunya.

## 2. State-Of-The-Art

**Semantic Segmentation** performs pixel-level classification, being CNNs the standard approach when annotated data is available. The widely adopted *U-Net* follows an encoder-decoder structure, gradually reducing and recovering spatial resolution. While other architectures (Zhao et al., 2017; Chen et al., 2017; Huang et al., 2019; Salpea et al., 2022) have shown strong results, U-Net remains dominant in biomedical imaging due to its simplicity and effectiveness (Isensee et al., 2018; Zhou et al., 2020). Meanwhile, specialized methods such as StarDist (Schmidt et al., 2018) and Cellpose (Stringer and Pachitariu, 2024) are particularly popular in fluorescence and immunofluorescence contexts, where they focus on capturing precise cell boundaries in often high-contrast images.

**Cell Counting** methods offer an alternative to cell segmentation by framing the task as density estimation. (Lempitsky and Zisserman, 2010) introduced a supervised framework that estimates object counts from dot annotations, bypassing explicit segmentation. (Xie et al., 2018) extended this with CNN-based fully convolutional regression networks for microscopy cell counting.

**Multi-Task Approaches** are a common strategy for instance segmentation in cell analysis, where semantic segmentation is combined with auxiliary tasks. HoVer-Net employs a

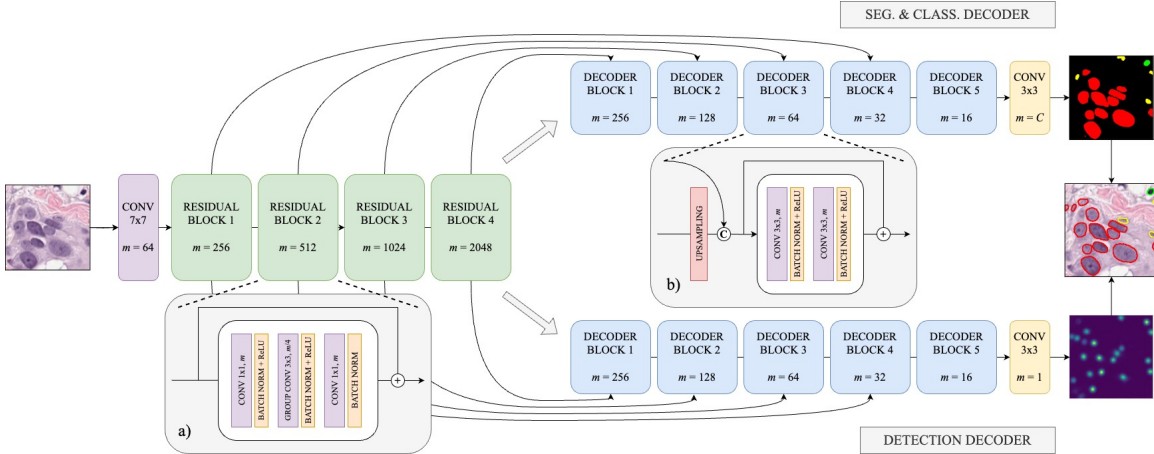

Figure 1: DualU-Net architecture. The encoder (green) extracts features and feeds two parallel decoders (blue): segmentation and classification (top) and detection (bottom). Each block outputs $m$ feature maps, with final heads (yellow) producing a multi-class segmentation mask ($m = C$) and a single-channel density map ($m = 1$). Insets (a) and (b) detail the residual and decoder block structures.

*three-headed* decoder and this design enables robust instance-level segmentation and classification. More recently, in our previous work (Anglada-Rotger et al., 2024), we introduced two independent U-Net models following a similar multi-task learning principle. Instead of HoVer maps, this model combines semantic segmentation with a cell counting task that estimates Gaussian-based cell centroids, effectively enabling cell separation even in highly overlapping regions.

**Transformer-based approaches** have recently gained traction in biomedical image analysis, with many adopting the three-headed scheme introduced by HoVer-Net (Graham et al., 2019). Examples of this strategy are CellViT (Hörst et al., 2024), which exemplifies a state-of-the-art Transformer architecture for cell segmentation; or NuLite (Tommasino et al., 2024), which prioritizes computational efficiency. Although Transformers capture long-range dependencies, they usually require more computational resources than CNN models. In contrast, CellDETR (Pina et al., 2024) uses the DETR framework to detect cells via bounding boxes rather than producing segmentation masks, making direct comparisons with HoVer-Net, CellViT, NuLite, or our method less applicable.

**Advanced Convolutional architectures** have been proposed integrating Transformer-inspired design principles. ConvNeXt (Liu et al., 2022) is one such architecture that rethinks standard ResNet-like backbones using modern components. ConvNeXt can serve as a drop-in replacement for earlier CNN backbones in tasks like U-Net. Its design also better aligns with multi-head decoder strategies by providing robust hierarchical feature representations.

## 3. Methods

In contrast to our earlier approach that used two independent U-Net models, we now unify both tasks—semantic segmentation and cell center detection—within a single network. The proposed DualU-Net architecture (Fig. 1) integrates two specialized decoder branches. While the encoder remains shared and captures multiscale features from the input images, each decoder targets a different objective: one for semantic segmentation and another for cell center detection.

**Two Heads Are Enough**  With our DualU-Net design, we aim to simplify the widely adopted three-decoder architecture model while addressing the same task (cell nuclei classification and instance segmentation and maintaining high performance, using only two decoder heads. First, we carefully weight the background class in the loss function by adjusting its importance through a tunable parameter to ensure balanced training despite the large background portion. This emphasizes the binary classification task of distinguishing cells from the background, enhancing segmentation accuracy. As a result, a dedicated Nuclei Pixel (NP) branch, as used in HoVer-Net, becomes redundant. Second, we estimate cell centroids using Gaussian-based density maps, predicting the center of mass of cells as Gaussian distributions. This approach provides a computationally efficient and interpretable method for cell detection. By adopting this strategy, we present a faster and more intuitive alternative to the HV vector branch in HoVer-Net.

**Encoder**  The encoder in the DualU-Net architecture is designed to extract multiscale feature representations from input images, leveraging the hierarchical structures of modern convolutional backbones. We tested two state-of-the-art architectures: ResNeXt-50 32×4d (Xie et al., 2016) and ConvNeXt-Base (Liu et al., 2022).

**Decoders**  The semantic segmentation decoder generates pixel-wise classification masks by progressively refining the feature maps across five hierarchical levels. The second decoder head predicts a Gaussian-based density map of cell centers. Ground-truth density maps are created using a Gaussian kernel over point annotations placed at each cell's centroid. During inference, local maxima on the predicted density map correspond to cell centers (Anglada-Rotger et al., 2024). The main architecture of the two decoders is the same, and it is represented in Fig. 1. However, while the semantic segmentation decoder final head maps the output to the required number of semantic classes, the detection decoder produces a single-channel density map, representing the likelihood of cell centers.

**Merging and Final Cell Instances**  In the final stage, the outputs from both decoder heads are merged to achieve instance-level segmentation. A watershed algorithm is applied to the semantic segmentation mask, using the predicted cell centers from the detection decoder as markers. Cells without an associated predicted center are discarded. This process effectively separates clustered cells, forming distinct connected components that correspond to individual cells. Each connected component is then assigned a semantic class through a majority vote based on the segmented pixels within it.

**Loss Function**  To train DualU-Net for its dual objectives, we employ a composite loss function that optimizes both tasks simultaneously. The total loss $\mathcal{L}_{\text{total}}$ is defined as:

$$\mathcal{L}_{\text{total}} = \lambda_{\text{dice}}\mathcal{L}_{\text{dice}} + \lambda_{\text{ce}}\mathcal{L}_{\text{ce}} + \lambda_{\text{mse}}\mathcal{L}_{\text{mse}}, \tag{1}$$

where $\lambda_{\text{dice}}$, $\lambda_{\text{ce}}$, and $\lambda_{\text{mse}}$ are weighting factors that control the contributions of the Dice loss $\mathcal{L}_{\text{dice}}$, the Cross-Entropy (CE) loss $\mathcal{L}_{\text{ce}}$, and the Mean Squared Error (MSE) loss $\mathcal{L}_{\text{mse}}$, respectively. The $\mathcal{L}_{\text{dice}}$ and $\mathcal{L}_{\text{ce}}$ loss primarily influence the segmentation task by ensuring accurate pixel-wise classification and mitigating class imbalance. Additionally, our experiments indicate that $\mathcal{L}_{\text{dice}}$ has a stronger influence on segmentation quality, whereas $\mathcal{L}_{\text{ce}}$ is more pivotal for improving classification performance. Relying on only one of these losses tends to focus the model on a single task and degrades performance on the other. Consequently, we adopt a combination of both, as also done in (Graham et al., 2019; Tommasino et al., 2024; Hörst et al., 2024). Furthermore, we conducted a hyperparameter search to tune the specific weighting factors and observed that giving them equal contribution ($\lambda_{\text{dice}} : \lambda_{\text{ce}} : \lambda_{\text{mse}} = 1 : 1 : 1$) yields the best overall performance. To ensure that underrepresented classes receive greater importance during training, we applied class-weighting strategies in which the loss contributions of each class, including the background, are weighted by the inverse of their frequency in the dataset. Meanwhile, the $\mathcal{L}_{\text{mse}}$ loss directly supervises the centroid estimation task by minimizing the error between the predicted Gaussian density map and the ground-truth center annotations. By enforcing a smooth and accurate density representation of cell centroids, this loss helps refine cell localization without requiring an explicit boundary prediction.

## 4. Results

**Evaluation Metrics**  We evaluated our model using metrics for detection, classification, and segmentation, following the definitions provided in the HoVer-Net (Graham et al., 2019) and PanNuke (Gamper et al., 2020) papers. For classification and detection, we used F1 scores. The detection F1 score ($F_{1,d}$) measures the accuracy of nucleus centroid localization, while the classification F1 score ($F_{1,c}$) evaluates the accuracy of cell type predictions. For segmentation, we primarily report the Dice coefficient. While Panoptic Quality (PQ) has been widely used in digital pathology, recent studies (Foucart et al., 2023) have demonstrated that PQ is unsuitable for cell nucleus instance segmentation and classification tasks. Despite these limitations, we include PQ for comparative purposes, as it remains a commonly reported metric.

**Experiments**  To evaluate the performance and efficiency of the DualU-Net, we conduct a series of experiments, including cell segmentation and classification benchmarking, inference time and computational efficiency analysis, and robustness assessment under staining variations. Detailed implementation settings for all experiments are provided in Appendix B.

**Cell Segmentation and Classification Results**  The performance of our models was evaluated on the PanNuke, CoNSeP, and Ki-67 datasets (see Appendix A) and compared to state-of-the-art approaches (see Table 1). On the PanNuke dataset, the $F_{1,d}$ of our ResNeXt-based model (0.80) and ConvNeXt-based model (0.80) is comparable to HoVer-Net (0.80), and closely follows NuLite-M (0.83) and CellViT (0.82). Regarding classification metrics, our models achieved equivalent $F_{1,c}$ for most categories while demonstrating superior performance in the less-represented Dead class (0.36) compared to HoVer-Net (0.31).

| Original | Ground Truth | Ours | Ours CN | HoVer-Net |
|----------|--------------|------|---------|-----------|

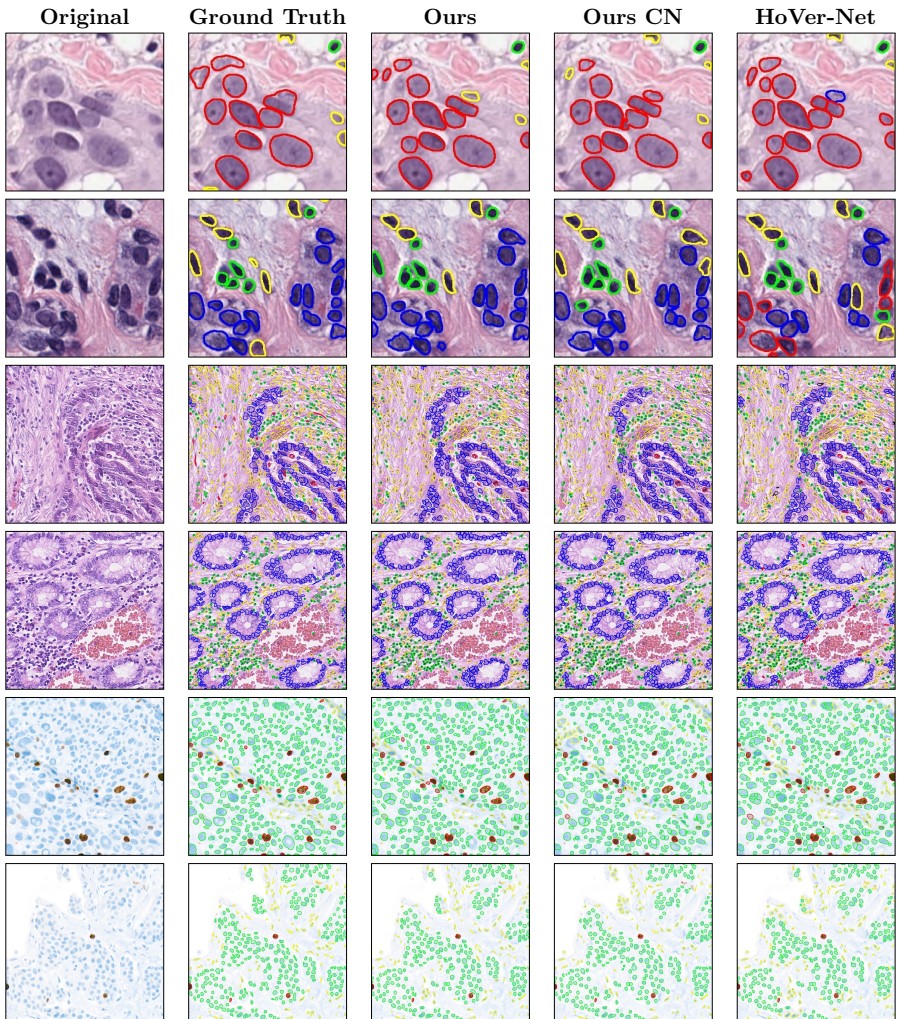

Figure 2: Qualitative results across PanNuke (rows 1–2), CoNSeP (3–4), and Ki-67 (5–6). "Ours" = ResNeXt-based model, "Ours CN" = ConvNeXt-based model. Overall, we observe no major differences in classification performance across the datasets, with notable improvements in PanNuke (particularly row 2). However, a slight downgrade in segmentation quality can be seen in cases like image 4 of row 1, reflecting typical watershed artifacts.

On the CoNSeP dataset, the $F_{1,d}$ of our models (0.72) is comparable to HoVer-Net(0.75). Classification performance for specific cell types, such as Epithelial (0.62) and Inflammatory (0.63–0.64), aligns with state-of-the-art results, while achieving superior results for the less-represented Miscellaneous class (0.44) compared to HoVer-Net (0.43).

For the Ki-67 dataset, the $F_{1,d}$ of our ResNeXt-based (0.80) and ConvNeXt-based (0.80) models are comparable to HoVer-Net (0.82). $F_{1,c}$ for Negative, Positive, and Stroma classes

Table 1: Performance across PanNuke, CoNSeP, and Ki-67 datasets. For PanNuke and Ki-67, the reported metrics represent the average across multiple dataset folds (3 and 4 respectively). The Dice metric for PanNuke and the mPQ for CoNSEP are not reported for state-of-the-art models, as they are not provided in the referenced papers. In Ki-67, HoVer-Net models were trained from scratch, and mPQ is not included due to the absence of an official implementation in its repository.

| | | Classification and Detection ↑ | | | | | | Segmentation ↑ | | |
|---|---|---|---|---|---|---|---|---|---|---|
| **Dataset** | **Model** | $F_{1,d}$ | $F_{1,c_1}$ | $F_{1,c_2}$ | $F_{1,c_3}$ | $F_{1,c_4}$ | $F_{1,c_5}$ | **Dice** | **mPQ** | **bPQ** |
| PanNuke | | | Neo. | Non-Neo. | Inflam. | Connect. | Dead | | | |
| | HoVer-Net (Gamper et al., 2020) | 0.80 | 0.62 | 0.56 | 0.54 | 0.49 | 0.31 | - | 0.46 | 0.66 |
| | CellViT$_{256}$ (Hörst et al., 2024) | 0.82 | 0.69 | 0.70 | 0.58 | 0.52 | 0.37 | - | 0.48 | 0.67 |
| | NuLite-M (Tommasino et al., 2024) | 0.83 | 0.70 | 0.73 | 0.58 | 0.52 | 0.37 | - | 0.50 | 0.68 |
| | Ours | 0.80 | 0.64 | 0.63 | 0.56 | 0.50 | 0.36 | 0.76 | 0.41 | 0.55 |
| | Ours ConvNeXt | 0.80 | 0.66 | 0.61 | 0.58 | 0.53 | 0.36 | 0.80 | 0.41 | 0.56 |
| CoNSeP | | | Epithelial | Inflammatory | Spindle | Misc. | | | | |
| | HoVer-Net (Graham et al., 2019) | 0.75 | 0.64 | 0.63 | 0.57 | 0.43 | | 0.85 | - | 0.52 |
| | Ours | 0.72 | 0.62 | 0.63 | 0.56 | 0.44 | | 0.77 | - | 0.34 |
| | Ours ConvNeXt | 0.72 | 0.62 | 0.64 | 0.57 | 0.34 | | 0.74 | - | 0.34 |
| Ki-67 | | | Negative | Positive | Stroma | | | | | |
| | HoVer-Net | 0.82 | 0.56 | 0.65 | 0.50 | | | 0.86 | - | 0.69 |
| | Ours | 0.80 | 0.54 | 0.66 | 0.43 | | | 0.83 | - | 0.62 |
| | Ours ConvNeXt | 0.80 | 0.57 | 0.66 | 0.47 | | | 0.83 | - | 0.63 |

show close agreement across all models, with our ConvNeXt-based model demonstrating a slight edge in the Negative class (0.57 vs. 0.54) and the Stroma class (0.47 vs. 0.43).

Regarding segmentation metrics, our models also achieved reasonable results: On the PanNuke dataset binary Panoptic Quality (bPQ) scores (0.55 and 0.56). In CoNSeP Dice (0.74 and 0.77) and bPQ (0.34 for both). In Ki-67 (0.83 Dice for both models), which is comparable to state-of-the-art models such as HoVer-Net (0.86).

A qualitative comparison of the results for our approaches and HoVer-Net across the three datasets is presented in Fig. 2, highlighting that segmentation results are qualitatively equivalent. Let us note that segmentation primarily serves a visualization role, while classification and detection remain the key factors for clinical decision-making.

**Inference Time and Computational Efficiency** It has been well-established that HoVer-Net is not optimal for fast and efficient processing (Baumann et al., 2024; Tommasino et al., 2024). Given this limitations, we compare our models to HoVer-Net in terms of inference time and also evaluate computational efficiency against state-of-the-art models: CellViT (performance) and NuLite (efficiency). Our models significantly reduce inference time compared to HoVer-Net. On the CoNSeP test set, our ResNeXt-based and ConvNeXt-based models complete inference in 66.3s and 65.8s, respectively, achieving a ×2.5 reduction over HoVer-Net (168.35s). On PanNuke, they process images in 108.1s and 137.6s, yielding up to a ×5.1 speed-up over HoVer-Net (551.45s). For a fair comparison, both codes were implemented in Python, we used the official HoVer-Net repository, and we did not extensively optimize the DualU-Net inference code. Our significantly lower runtime arises from

Table 2: Performance comparison of different models for input sizes 256x256 and 1024x1024. Results are extracted from (Tommasino et al., 2024).

| Model | Nº Parameters (M) | GLOPs ↓ | | Latency (ms) ↓ | |
|---|---|---|---|---|---|
| | | 256 | 1024 | 256 | 1024 |
| CellViT$_{256}$ | 46.75 | 132.89 | 2125.94 | $35.71 \pm 0.37$ | $1169.7 \pm 148.92$ |
| NuLite-S | 34.10 | 23.15 | 370.25 | $29.99 \pm 1.79$ | $310.44 \pm 24.64$ |
| NuLite-M | 47.93 | 32.54 | 520.45 | $33.37 \pm 1.34$ | $446.3 \pm 35.25$ |
| Ours | 41.01 | 16.26 | 260.23 | $12.05 \pm 0.41$ | $141.88 \pm 0.69$ |
| Ours ConvNeXt | 97.81 | 26.78 | 428.49 | $20.82 \pm 0.17$ | $264.19 \pm 1.48$ |

i) having only two decoder branches instead of three, ii) avoiding HV vector predictions, and iii) generating instance boundaries via watershed from centroid maps.

Despite having more parameters, our models improves computational efficiency. Our ResNeXt-based model surpasses NuLite-S, as shown in Table 2, achieving lower GLOPs (30% lower for $1024 \times 1024$ images) and a significantly reduced latency. Our ConvNeXt-based model, despite its higher parameter count, remains competitive, requiring fewer GLOPs than NuLite-M and achieving latency close to NuLite-S. These improvements highlight the efficiency of our approach in reducing computational overhead without sacrificing performance.

**Robustness to Color Variations** Histopathological WSIs often exhibit color variations due to inconsistencies in staining protocols and scanning conditions, which can affect model performance. To evaluate the robustness of our models to these variations, we generated five augmented versions of the CoNSeP test dataset. This augmentation involved random 90-degree rotations, flips, and perturbations in the Hematoxylin-Eosin-DAB (HED) color space. This introduces realistic staining variations, enabling a more comprehensive assessment of model stability (see Appendix D for examples and generation details). The evaluation was conducted for our main approach, the ResNeXt-based model. It demonstrates 49.1% lower variance in $F_{1,d}$ ($0.70 \pm 0.0086$) compared to HoVer-Net ($0.73 \pm 0.0169$), suggesting greater consistency under varying staining conditions. Similarly, the mean $F_{1,c}$ of our model ($0.49 \pm 0.0367$) exhibits 15.6% reduced variance compared to HoVer-Net ($0.50 \pm 0.0435$). In segmentation, our model also shows 45.3% lower variance in Dice score ($0.75 \pm 0.0093$) compared to HoVer-Net ($0.82 \pm 0.0170$), further highlighting its robustness.

## 5. Discussion and Conclusions

This study introduces DualU-Net, a streamlined architecture for cell classification and segmentation in histopathology, developed to handle multiple staining protocols, including H&E and Ki-67. Our goal is to demonstrate that *Two Heads are Enough*, challenging the necessity of HoVer-Net's three-decoder paradigm, yet still tackling the same overall task of cell nuclei classification and instance segmentation. Although state-of-the-art models have widely adopted a three-decoder setup, DualU-Net consolidates its functionality by carefully weighting the background class in the loss function and adopting Gaussian-based density maps for centroid estimation. This makes the NP branch redundant and provides a faster, more intuitive alternative to HoVer-Net's HV representation.

Our results show that DualU-Net achieves comparable classification and detection performance to state-of-the-art models across multiple stains, while reducing architectural complexity, improving computational efficiency, and increasing robustness to color variations. Although slightly lower segmentation scores have been observed, they can be attributed to two factors (more details in Appendix C):

(i) Watershed-based segmentation, where our centroid-based approach, unlike boundary-focused methods, occasionally leads to non-smooth or irregular contours due to the inherent nature of the watershed algorithm (see Fig. 3, bottom).

(ii) Ground truth inconsistencies (see Fig. 3, top), notably oversegmentation in CoNSeP and missing cell annotations in PanNuke, directly affect the learning process of our center detection head by introducing errors in the Gaussian map generation (the foundation of our watershed algorithm). Consequently, these issues have a stronger impact on our segmentation metrics than approaches not driven by centroid-based segmentation.

Moreover, since segmentation is primarily a visualization tool, the qualitative results shown in Fig. 2 confirm for this aim an equivalent performance to state-of-the-art methods. Finally, our results indicate that ConvNeXt does not provide significant improvements over ResNeXt, reinforcing the efficiency of the original backbone.

DualU-Net significantly reduces inference time compared to HoVer-Net, making it more practical for real-world deployment. On CoNSeP, we process images $\times 2.5$ faster, and on PanNuke, we achieve a $\times 5.1$ speed-up. Additionally, DualU-Net is more computationally efficient than CellViT and NuLite. Despite having more parameters, it surpasses NuLite-S in efficiency. These improvements highlight the effectiveness of our approach in reducing computational complexity without sacrificing segmentation and classification accuracy.

Stain variations present a well-known challenge in histopathology, as differences in staining protocols and scanning devices can significantly impact model performance. Our controlled color perturbation experiments on CoNSeP confirm that DualU-Net exhibits lower variance in classification, detection and segmentation scores compared to HoVer-Net.

In conclusion, DualU-Net eliminates the need for a third decoder head, achieving classification and detection performance comparable to state-of-the-art models, along with competitive segmentation, while enhancing inference efficiency and robustness to color variations. These advantages make it well-suited for clinical deployment, where speed and efficiency are crucial. Furthermore, DualU-Net has been successfully integrated into the DigiPatICS project (Temprana-Salvador et al., 2022) and deployed in eight hospitals within the Institut Català de la Salut de Catalunya, highlighting its real-world impact. Future work will focus on exploring lighter models, such as ConvNeXt-Tiny (Liu et al., 2022), to further enhance computational efficiency.

## Acknowledgments

This publication is part of the R&D&I project PID2023-148614OB-I00, funded by MICIU/AEI/10.13039/501100011033/ and by FEDER, EU. This research has also been funded by European Development Funds Regional, Programa operatiu FEDER de Catalunya 2014-2020 through the project DigiPatICS.

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

## Appendix A. Datasets

**PanNuke** The PanNuke dataset (Gamper et al., 2020) is a large-scale collection of H&E stained histopathology images derived from 19 tissue types. It comprises 7,904 patches, each sized $256 \times 256$ pixels, extracted from WSIs from The Cancer Genome Atlas (TCGA) at a magnification of $40\times$. Within this dataset, there are 189,744 labeled nuclei classified into five classes: neoplastic, inflammatory, connective, necrosis, and epithelial.

**CoNSeP** The CoNSeP dataset (Graham et al., 2019) focuses on H&E colorectal adenocarcinoma samples. It comprises 41 patches, each 1000x1000 pixels in size, extracted from WSI at a magnification of 40×. The dataset encompasses various regions such as stromal, glandular, muscular, collagen, adipose, and tumorous areas and its nuclei are grouped into five classes: inflammatory, epithelial, spindle-shaped and miscellaneous.

**Ki-67** Additionally, we employ a custom Ki-67 dataset (Anglada-Rotger et al., 2024), developed within the DigiPatICS project (Temprana-Salvador et al., 2022), comprising 52 annotated tiles (each of size $1024 \times 1024$ pixels) extracted from Ki-67-stained WSIs at a magnification of 40×. Sourced from four patients exhibiting different proliferation levels, each tile is accompanied by cell-level annotations that include segmentation masks and cell classes (positive, negative, or non-epithelial). This dataset is not publicly available, and the weights of the models trained on it will not be released.

## Appendix B. Implementation Details

The models were trained using Adam Optimizer with a base learning rate of 0.0001 for 256×256 images and 0.002 for 1024×1024 images. Training was performed for 100 epochs with a weight decay of 0.0001, and the learning rate was reduced by a factor of 0.1 at epochs 70 and 90. As commented in Section 3, a hyperparameter search determined that equal contributions from all loss components yielded the best results, so all loss weights ($\lambda_{\text{dice}}$, $\lambda_{\text{ce}}$, and $\lambda_{\text{mse}}$) were set to 1. Gaussian density maps for cell centroid estimation were generated using a fixed standard deviation of $\sigma = 5$.

For the PanNuke dataset, the best checkpoint was selected based on the detection and classification metrics on the validation fold. In contrast, for the Ki-67 and CoNSeP datasets, where validation sets are unavailable, the final model at the 100th epoch was used. The models were trained on 2 NVIDIA GeForce RTX 3090 GPUs (24 GB each), using a batch size of 4 per GPU for 1024×1024 images and 8 per GPU for 256×256 images. For data augmentation, horizontal and vertical flips and 90-degree rotations were applied, each with a probability of $p = 0.5$.

## Appendix C. Analysis of Segmentation Performance and Ground Truth Limitations

As commented in Section 5, our watershed-based segmentation relies on centroid predictions (density maps) rather than explicit boundary features. Although this approach accelerates and simplifies the pipeline, particularly in classification-focused tasks, it can sometimes yield irregular boundaries or non-smooth contours when nuclei are tightly clustered or when centroids are poorly localized. Even minor errors in centroid placement can propagate through the watershed algorithm, resulting in segmentation artifacts (see Figure 3, bottom). In addition, the ground truth annotations in certain datasets introduce further challenges. CoNSeP includes instances of oversegmented annotations that can artificially inflate metrics for methods matching those finer divisions. Conversely, PanNuke has missing nuclei, penalizing models that detect unlabeled cells.

Although these issues can negatively affect our reported segmentation scores, they do not undermine the main goal of DualU-Net, which is to deliver accurate cell detection and

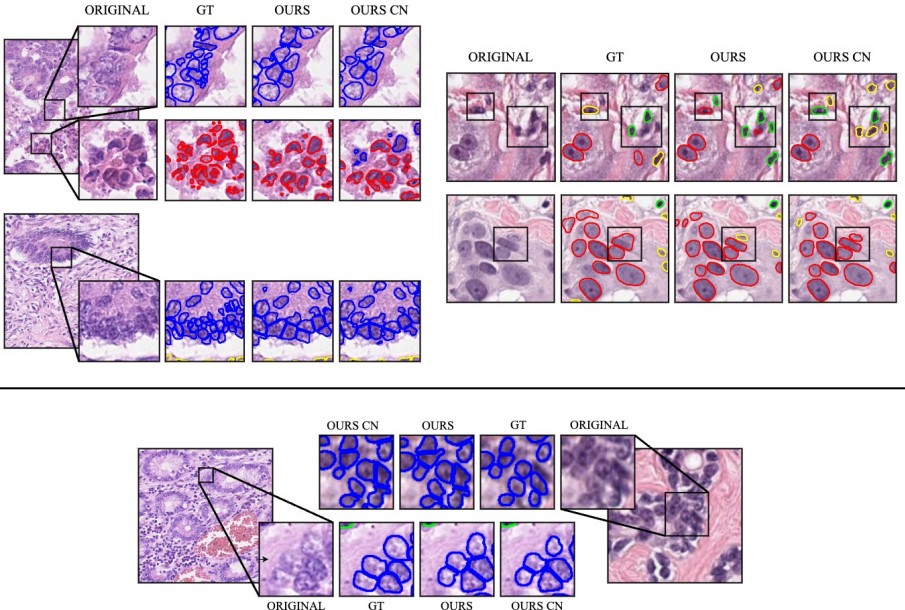

Figure 3: Top: Examples of ground truth annotation inconsistencies in the datasets. On the left (CoNSeP), the ground truth exhibits supersegmentation. On the right (PanNuke), missing cell annotations are observed. Bottom: Examples of segmentation artifacts introduced by the watershed algorithm. On the left, CoNSeP, and on the right, PanNuke.

classification. This emphasis aligns well with real-world pathology workflows, where precise cell counts and classifications typically carry more clinical significance than perfectly smooth nucleus boundaries.

## Appendix D. Visualization of Color Perturbations

To qualitatively assess the impact of staining variations, Figure 4 presents sample images from the CoNSeP test set alongside their perturbed versions and corresponding model predictions. These variations were introduced using the HEDJitter transformation (Tellez et al., 2018; Ruifrok and Johnston, 2001), which modifies the Hematoxylin (H) and Eosin (E) channels in the HED color space before converting the image back to RGB. Specifically, the intensity of each channel was scaled by a random factor $\alpha \sim U(0.98, 1.02)$ and shifted by a bias $\beta \sim U(-0.02, 0.02)$, mimicking real-world staining inconsistencies encountered in histopathology slides.

Figure 4 showcases how these color perturbations affect visual appearance while maintaining structural integrity, allowing us to evaluate model robustness against staining-induced variations. Predictions from our model and HoVer-Net are provided for comparison. This visualization complements the quantitative results in Section 4, reinforcing the stability of our approach under varying staining conditions.

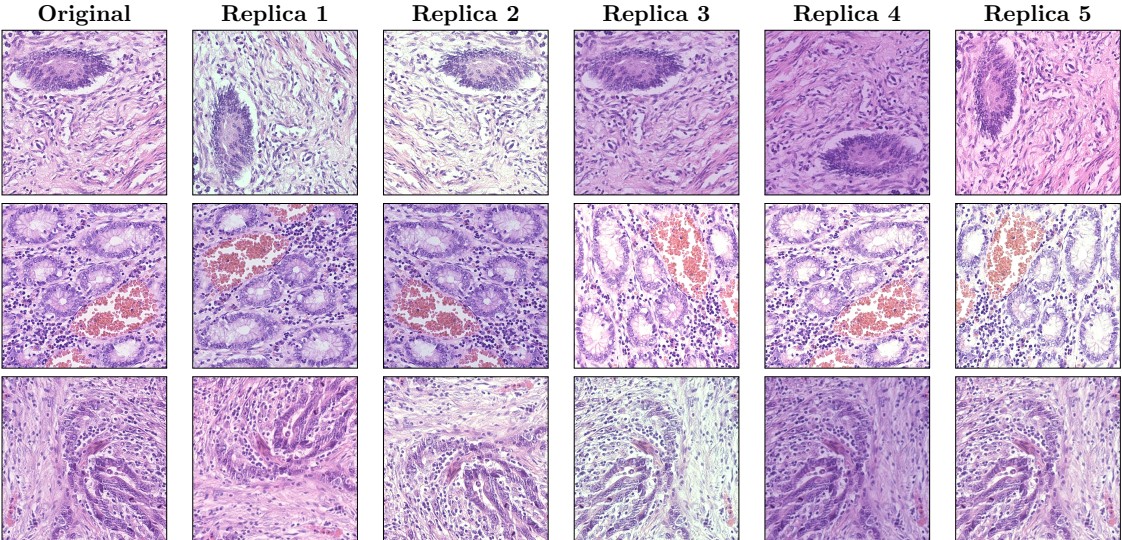

Figure 4: Examples of color perturbations applied to the CoNSeP test set. The first column presents the original images, while the remaining columns display five perturbed replicas generated.

