# OpenReview forum: "Two Heads Are Enough: DualU-Net, a Fast and Efficient Architecture for Nuclei Instance Segmentation"
_MIDL.io/2025/Conference — MIDL 2025 Oral_

### Official Review · Reviewer_ubvp · 2025-02-19

**Confidence:** 4
**Preliminary Rating:** 4
**Recommendation:** Oral
**Final Rating:** 4

**Summary:**

The paper proposes an approach to overcome the limitations of the HoverNet architecture, particularly to reduce the computational complexity. Much improvement is reported in inference time for the experiments conducted on the three datasets.

**Strengths:**

A key contribution of this work is the real-world deployment as the paper has claimed the integration of the proposed DualU-Net architecture into the DigiPatICS project and a deployment across eight hospitals.
Experiments are well reported. Evaluation is done using a variety of metrics for classification and segmentation.

**Weaknesses:**

The paper claims to reduce computational overhead by presenting an alternative to Hover-net that has a three-decoder architecture. However, this is not a fair comparison as the Hover-net used a three-decoder architecture to achieve three tasks. Here, the authors are addressing two tasks i.e., segmentation and classification tasks only.

**Detailed Comments:**

As listed above.

**Justification Of The Final Rating:**

The authors have addressed my concerns. I have read the responses and based on these, I would be happy to see the paper accepted in the conference. However, I am retaining my previous voting as I have been generous already with this.

**Justification Of The Preliminary Rating:**

I would be happy to see this work accepted. I believe there is value in the work. Though there are areas that could have been improved, the authors have done a decent job in presenting the concepts and the experiments.

**Questions To Address In The Rebuttal:**

I would have expected a more detailed caption for Figure 2 to give the reader a clear understanding of what differences the authors are trying to identify in these images.

**Special Issue:**

No

---

> ### Author Response · Authors · 2025-03-07
>
> Thank you for your detailed feedback and for your positive remarks about our real-world deployment and experimental design. Below, we address your main points:
>
> ## 1. Figure 2 Caption
> We acknowledge that the caption in the original submission was too concise. We have now expanded it to give the reader a clearer sense of the classification results (which compare favorably to the state-of-the-art) and discuss some segmentation artifacts. These changes clarify more precisely which differences we aim to illustrate in the figure.
>
> ## 2. Comparison with HoVer-Net
> We understand your concern that HoVer-Net uses three decoders for three tasks, whereas we only use two. However, our view is that the tasks tackled (cell nuclei segmentation and classification) align with HoVer-Net’s primary goals—even if HoVer-Net’s third branch is a dedicated binary segmentation decoder. One of our central contributions is demonstrating that this third head is largely redundant: We handle binary segmentation by carefully weighting the background class in the same decoder that performs multi-class segmentation, and we replace the HoVer “HV” branch with a more efficient and intuitive centroid-detection approach. Thus, the overall objectives remain comparable (classify and segment cell nuclei), and we believe the direct comparison is appropriate. We have made this rationale clearer in Section 3 (“Two Heads are Enough” paragraph) and Section 5 of the revised text.
>
> We appreciate your recommendation and the helpful suggestions for improvement. We believe these clarifications and updates strengthen our case that DualU-Net is a more streamlined solution to the same problem space HoVer-Net operates in. Thank you again for your time.

---

> > ### Comment · Reviewer_ubvp · 2025-03-10
> >
> > Thank you for providing this explanation.

---

### Official Review · Reviewer_J8Wq · 2025-02-21

**Confidence:** 5
**Preliminary Rating:** 5
**Recommendation:** Poster
**Final Rating:** 5

**Summary:**

This paper proposes DualU-Net, which uses a new schema for post-processing network raw inference into cell instances. In the new framework, the model predicts two types of outputs: semantic segmentation map and a centroid density map. Subsequently, individual cell instances are obtained by using watershed on the foreground region with local maxima in the density map as seeds.

**Strengths:**

- The proposed instance representation schema makes sense for nucleus segmentation. Using the local maxima to find instance centroid also avoids the need for making multiple candidate predictions and using non-maxima suppression like in StarDist.
- In some sense I think the isotropic nature of Gaussian representation is easier for the model to predict.
- The claimed speed-up is attractive compared to HoVerNet.

**Weaknesses:**

- The shape of instances the proposed schema can produce is limited by the watershed operation. Consider a scenario where a round nucleus is hugging an elongated nucleus around its perimeter. The model will not be able to handle this case very well, while schema such as HoVerNet and Cellpose can in theory can.
- In all fairness, such cases are rather rare for nucleus segmentation but can be very common for cell segmentation with immunofluorescence images.
- Relying on watershed can also lead to unnatural shapes for nuclei, such as one of the red nuclei in Fig 2 row 1 image 4.

**Detailed Comments:**

- The authors should also extend their experimental comparisons to methods such as cellpose and stardist.
- The authors should also provide some insights on how much of the speed-up is brought by better algorithm design vs optimized implementation. For example, the same routine implemented in Python is generally much slower than in C.

**Justification Of The Final Rating:**

Retaining my preliminary decision:

The proposed instance representation schema makes sense for nucleus segmentation. Using the local maxima to find instance centroid also avoids the need for making multiple candidate predictions and using non-maxima suppression like in StarDist.
In some sense I think the isotropic nature of Gaussian representation is easier for the model to predict.
The claimed speed-up is attractive compared to HoVerNet.

**Justification Of The Preliminary Rating:**

- The proposed instance representation schema makes sense for nucleus segmentation. Using the local maxima to find instance centroid also avoids the need for making multiple candidate predictions and using non-maxima suppression like in StarDist.
- In some sense I think the isotropic nature of Gaussian representation is easier for the model to predict.
- The claimed speed-up is attractive compared to HoVerNet.

**Questions To Address In The Rebuttal:**

See above.

---

> ### Author Response · Authors · 2025-03-07
>
> Thank you for your detailed and constructive feedback. We appreciate your positive assessment of DualU-Net’s instance representation schema and the speed-up we achieve relative to HoVer-Net. Below we address your key points and clarify several aspects of our study:
>
> ## 1. Inference Time and Implementation Details
> We agree that comparing algorithmic versus implementation-level speed-ups is crucial. In our experiments, we used the official Python implementation of HoVer-Net and did not extensively optimize the Python inference code of DualU-Net. This helped ensure a fair, like-for-like comparison. These details have been added to the revised text in Section 4. Additionally, the computational efficiency figures reported for CellViT and NuLite in Table 2 in our paper were extracted from Tommasino et al. (2024). While further low-level optimizations (e.g., in C) might improve raw runtimes for both HoVer-Net and DualU-Net, the overall relative improvement we observed stems mainly from our more compact, two-decoder design.
>
> ## 2. Comparisons to Cellpose and StarDist
> We appreciate your suggestion to compare DualU-Net against Cellpose and StarDist. We are aware of both methods and their strong performance—especially for cell segmentation in fluorescence or immunofluorescence images. However, our core baselines (HoVer-Net, CellViT, NuLite) focus heavily on multi-task learning for classification and segmentation in brightfield (H&E-like) contexts; moreover, none of these baseline papers included Cellpose or StarDist in their own comparative studies, which further guided our choice of baselines.
> Furthermore, Cellpose and StarDist place more emphasis on intricate instance boundaries (often for purely segmentation-driven tasks), whereas our architecture and experiments are primarily geared toward classification, using segmentation only as a visualization or approximate delineation tool. That said, we agree that these methods offer valuable insights, and we do plan to evaluate DualU-Net against them in future studies—particularly for scenarios where instance shape is the paramount concern. We have also added references to Cellpose and StarDist in Section 2 (“Semantic segmentation” paragraph) to acknowledge their relevance.
>
> ## 3. Watershed Limitations
> We acknowledge that watershed may produce less natural contours when nuclei are tightly adjacent or elongated, which can sometimes happen in immunofluorescence or specialized staining protocols. We have expanded Sections 5 and Appendix C to explicitly comment these potential drawbacks, as well as our motivation to accept a slight reduction in segmentation smoothness in exchange for faster inference and robust classification. Since classification and cell counting are typically the top clinical priorities, our approach prioritizes their accuracy while using segmentation as a secondary, visualization-focused feature.
>
> Once again, thank you for your encouraging review and for highlighting these important points. We believe the revised text clarifies our choices regarding efficiency comparisons, the scope of our baselines, and the trade-offs associated with watershed-based segmentation. We sincerely appreciate your recommendation and look forward to any further feedback.

---

> ### Comment · Area_Chair_ePU9 · 2025-03-14
>
> Dear Reviewer,
>
> The discussion period ends in less than 24 hours, and your final rating is missing. Please review the authors’ response, revisions, and peer feedback, then update your score. You can submit your final rating along with the justification by editing your original review.
>
> Your final rating is critical for the decision.
>
> Thanks,
> AC, MIDL 2025

---

### Official Review · Reviewer_jEH5 · 2025-02-28

**Confidence:** 4
**Preliminary Rating:** 5
**Recommendation:** Oral
**Final Rating:** 5

**Summary:**

The paper introduces DualU-Net, a novel multi-task deep learning architecture for cell nuclei classification and segmentation in histopathological images. Unlike the widely used HoVer-Net, which relies on three decoders, DualU-Net employs only two output heads: one for segmentation and another for centroid detection. This design results in a 5x faster inference time compared to HoVer-Net while maintaining comparable accuracy in both classification and detection tasks. Additionally, DualU-Net demonstrates greater computational efficiency and enhanced robustness to staining variations, which is critical for real-world pathology applications. The model's deployment across eight hospitals as part of the DigiPatICS initiative highlights its practical viability in clinical environments.

**Strengths:**

- Computational Efficiency and Speed: DualU-Net reduces inference time by up to 5x compared to HoVer-Net, enabling real-time deployment in clinical settings without compromising accuracy.
- Novel Architecture Design: The paper challenges the conventional three-decoder paradigm by proving that two heads (segmentation and detection) are sufficient for effective cell nuclei classification and segmentation.
- Robustness to Staining Variations: The model maintains consistent performance across different histological stains, addressing a significant challenge in digital pathology workflows.

**Weaknesses:**

- Slightly Lower Segmentation Accuracy: The segmentation performance, particularly in contour smoothness, is slightly lower compared to HoVer-Net, attributed to the watershed algorithm used for instance-level segmentation.
- Ground Truth Inconsistencies: The datasets used (PanNuke and CoNSeP) show annotation inconsistencies, such as supersegmentation and missing cell labels, which could affect model evaluation metrics. N.B. This is a potential weakness of this study setting, but NOT a weakness of this method proposed.

**Detailed Comments:**

See Questions To Address In The Rebuttal Section

**Justification Of The Final Rating:**

The response (along with its corresponding revisions) from the author has resolved all my concerns. Therefore, I remain to support the acceptance of this article. I re-emphasise the key highlights of this article as below.

The paper introduces DualU-Net, a novel multi-task deep learning architecture for cell nuclei classification and segmentation in histopathological images. Unlike the widely used HoVer-Net, which relies on three decoders, DualU-Net employs only two output heads: one for segmentation and another for centroid detection. This design results in a 5x faster inference time compared to HoVer-Net while maintaining comparable accuracy in both classification and detection tasks. Additionally, DualU-Net demonstrates greater computational efficiency and enhanced robustness to staining variations, which is critical for real-world pathology applications. The model's deployment across eight hospitals as part of the DigiPatICS initiative highlights its practical viability in clinical environments.

**Justification Of The Preliminary Rating:**

The paper makes a significant contribution to digital pathology by introducing a more efficient and robust architecture, DualU-Net. The performance gains in computational efficiency and robustness to staining variations justify its acceptance. However, the slightly lower segmentation accuracy and certain architectural choices warrant further clarification. Nevertheless, I would recommend acceptance of this submission highly.

**Questions To Address In The Rebuttal:**

- Ground Truth Inconsistencies: How do the annotation inconsistencies in PanNuke and CoNSeP datasets impact the reported performance metrics, especially segmentation accuracy?
- Loss Function Tuning: Could the authors provide additional details on the impact of different weighting factors in the composite loss function?

**Special Issue:**

Yes

---

> ### Author Response · Authors · 2025-03-07
>
> Thank you for your constructive and positive review. Below is a concise summary of our revisions, along with a small ablation study we performed specifically for the CoNSeP dataset:
>
> ## 1. Ground Truth Inconsistencies:
> We have expanded Sections 5 and Appendix C to describe how missing annotations in PanNuke and oversegmentation in CoNSeP can disproportionately penalize our centroid-based watershed approach. In particular, faulty or inconsistent cell centers in the ground truth can introduce errors in our Gaussian map generation, which directly affects segmentation performance when using watershed. We emphasize that such misalignments often misrepresent the true efficacy of our detection and classification stages, both of which remain robust.
>
> ## 2. Loss Function Tuning and Ablation on CoNSeP
> We have clarified in Section 3 that Dice loss mainly boosters segmentation, while Cross-Entropy better boosts classification performance. We conducted a small ablation study specifically on the CoNSeP dataset. This study is not included in the revised text, but we share its highlights here to clarify our reasoning:
> - *Dice Only*: Dice Score of 0.74, Detection F1 of 0.68, and a mean Classification F1 of 0.28.
> - *Cross-Entropy Only*: Dice Score of 0.63, Detection F1 of 0.63, and a mean Classification F1 of 0.44.
> - *Combined Dice + Cross-Entropy*: Balanced improvements in all metrics (as shown in our paper’s main results).
>
> These observations underline why adopting both Dice and Cross-Entropy is needed. Relying on a single loss skews performance either toward segmentation or classification, whereas combining them finds a better sweet spot.
>
> ## 3. Textual Revisions
> - In Section 5 and Appendix C, we describe how centroid misannotations propagate more acutely in our method than in boundary-based approaches.
> - In Section 3 (“Loss function” paragraph), we elaborate on how the synergy of Dice and Cross-Entropy ensures that we do not overemphasize one aspect of the model’s output at the expense of the other.
>
> We appreciate that you found DualU-Net’s efficiency and robustness compelling. We hope these additional details and the small ablation study further elucidate why we chose our current loss function design, as well as how ground truth inconsistencies can affect centroid-based segmentation. Thank you again for your recommendation and insightful comments.

---

### Author Rebuttal · Authors · 2025-03-07

**Rebuttal:**

We attach the revised version of our manuscript, which has been thoroughly updated in response to the reviewers’ constructive comments. We sincerely thank all the reviewers for their positive feedback and suggestions, which have helped us improve both the clarity and rigor of our work. In this revision, all new or substantially revised text is highlighted in green so that readers can easily identify the changes and follow our updates. We believe that these enhancements address the key concerns raised during the review process and further strengthen our paper.

**Supporting Material:**

/attachment/03f2b2ff43f6955178d01feea156eeaaa59504d9.pdf

---

### Meta-Review · Area_Chair_ePU9 · 2025-03-23

**Recommendation:** Accept (Oral)
**Confidence:** 5

**Metareview:**

This paper received two strong accepts and one weak accept, with all reviewers recognizing its technical merit and practical impact. The proposed DualU-Net achieves substantial speedup over HoVer-Net while maintaining competitive performance, making it highly suitable for clinical deployment. Based on the positive evaluations and thoughtful rebuttals, I recommend acceptance.